# Effects of Essential Oil Fumigation on Potato Sprouting at Room-Temperature Storage

**DOI:** 10.3390/plants11223109

**Published:** 2022-11-15

**Authors:** Jena L. Thoma, Charles L. Cantrell, Valtcho D. Zheljazkov

**Affiliations:** 1Department of Crop and Soil Science, Oregon State University, Corvallis, OR 97331, USA; 2Natural Products Utilization Research Unit, Agricultural Research Service, United States Department of Agriculture, University City, MS 38677, USA

**Keywords:** potato storage, essential oils, sprout suppression, organic agriculture, room temperature

## Abstract

As a global staple, potato plays an important role in meeting human dietary needs and alleviating malnutrition. Potato sprouting during storage is a major issue that threatens food security by increasing food waste and must therefore be controlled. Biopesticides, including essential oils (EOs), have a history of use as potato sprout suppressants, and interest in their use has been renewed in response to stricter regulations on CIPC, the dominant chemical sprout suppressant over the last half-century. We evaluated twenty-one EOs as potential sprout suppressants in cv. Ranger Russet potatoes at room-temperature storage. Treatment with *Artemisia herba-alba* EO was the most effective at suppressing both sprout length and sprout number over a 90-day storage period. GC—MS—FID analysis of *A. herba-alba* EO revealed the presence of α-thujone, hexadecenoic acid, β-thujone, camphor, sabinene, and camphene at amounts >1%. *Cistus ladanifer*, *Ocimum basilicum*, *Ormenis mixta*, and *Salvia sclarea* EOs significantly reduced sprout length for shorter storage periods, whereas *Cinnamomum zeylanicum* (bark) and *Laurus nobilis* EOs also significantly reduced sprout number. *Syzygium aromaticum* (clove) EO did not significantly suppress sprouting at room temperature. These results indicate the potential of certain EOs to be used as sprout suppressants for room-temperature potato storage, providing needed alternatives for both organic and conventional potato industries.

## 1. Introduction

Ranking fourth in global crop production, potato (*Solanum tuberosum*) is of major importance to the world economy and global food security [1,2]. Of the 370 million tons of potato produced in 2019, 19 million tons were grown in the United States, ranked fifth in global production [3,4]. Sales of US-grown potatoes alone totaled over USD 3.6 billion in 2020 [5]. Cultivated in over 100 counties, fresh and processed potato products are highly popular and potato consumption plays an important role in reducing world hunger [6,7].

As a world staple, appropriate potato storage for many months is crucial to ensure adequate supplies to meet demands for both consumption and seed [8]. When potatoes are harvested, the many buds on their surfaces are in a natural state of dormancy and will not sprout [9]. However, this natural dormancy is transient, and the buds will begin to sprout after several months of storage, although the length of the dormant period varies between cultivars [10,11]. Potato sprouting during storage must be limited as sprouting leads to decreased tuber weight, changes in tuber texture and nutritional value, and the formation of solanine, an alkaloid that is toxic to mammals [9,12,13]. Due to solanine accumulation, sprouting potatoes are considered inedible and become food waste [7].

Physical and chemical means of sprout suppression are often employed to reduce potato sprouting during storage. For example, storage at temperatures between 8–12 °C at 85–90% relative humidity is the most common method of preserving processing potato quality for up to 9 months [8]. However, these temperatures are unable to completely suppress sprouting after dormancy is broken [8]. Lower temperatures could achieve better sprout suppression, but colder temperature storage causes unacceptable changes in processed product color and potential health risks when these potatoes are fried due to increased glucose concentrations and acrylamide formation [14,15]. Furthermore, cold-temperature storage may not be an option for small and medium facility owners that lack the capital to install and run such storage chambers [16]. The application of chemical sprout suppressants, on the other hand, is an effective and potentially low-cost method of sprout suppression.

Chlorpropham (CIPC) is a highly efficacious and inexpensive sprout inhibitor and has been the dominant chemical sprout suppressant since the mid-20th century [1]. However, CIPC and its metabolites have been linked to potentially adverse health and environmental effects, resulting in bans on its use in the European Union and restrictions on allowable residues in other countries including the US [17,18,19]. In 2017, US potato exports were valued at USD 3 billion [20]. With final use of CIPC products in the EU concluding in 2020, zero-tolerance policies for CIPC residues in countries around the world could negatively impact American potato exports [17,21]. Additionally, the world has witnessed growing global demand for organically produced foods and products; organic sales increased by 12.4% from 2020 to 2021, reaching USD 62 billion in the US alone in 2021 [22]. The organic market therefore represents a unique opportunity that could counteract potential revenue losses in the potato industry in response to stricter regulations on CIPC, but only if effective and organic sprout suppressants are identified, EPA registered, and their use scaled up.

There are several organic sprout suppressants currently available, including some essential-oil-containing products. For example, Biox-M, Biox-C, and Talent^®^ contain 100% spearmint (*Mentha spicata* L.), clove (*Syzygium aromaticum* L.), and caraway (*Carum carvi* L.) essential oils (EO), respectively, and all are marketed for potato sprout suppression [1]. The effectiveness of these products on sprouting varies with storage temperature, cultivar, and application scheme, presenting significant challenges to their widespread use in potato storage [16,21,23,24,25]. Furthermore, due to the wide variety of compounds present in EOs both within and among species, it stands to reason that there may be species that have sprout suppressive capabilities that have yet to be identified [26]. The goal of the present study was to evaluate the sprout-suppressive qualities of previously untested EOs in a single potato cultivar stored at room temperature, with the objective of identifying new, promising sprout suppressants.

## 2. Results and Discussion

### 2.1. Effects of Essential Oils (EOs) on Sprout Length

After 90 days of storage, a statistically significant two-way interaction between treatment and time was observed on sprout length (*p* < 0.001). This suggests that the impact of treatment on sprout length depends on the amount of time that has passed. Furthermore, the main effects of both treatment and time were significant (*p* < 0.001).

The EOs of *Ormenis mixta*, *Cinnamomum zeylanicum* bark, *Cistus ladanifer*, *Salvia sclarea*, *Artemisia herba-alba*, *Ocimum basilicum*, and *Laurus Nobilis* resulted in significant differences in sprout length relative to the control (Table 1). Treatment with *A. herba-alba* EO resulted in significant differences in sprout length from the control at all time points. *S. sclarea* EO treatment caused significant differences in sprout length up to 75 days. Sprout length in response to *C. ladanifer* and *L. nobilis* EOs significantly differed from the control for up to 60 days, whereas *C. zeylanicum* bark and *O. basilicum* EO treatments caused differences in sprout length for up to 45 and 30 days, respectively (Table 1).

None of the EOs completely suppressed sprouting, although *A. herba-alba* EO treatment limited average sprout length to less than 1 mm for up to 75 days and was the most effective inhibitor of sprout length throughout the 90-day storage period (Table 2). Sprout length due to *A. herba-alba* EO treatment differed significantly from all other treatments from 60 days until the end of the storage period (*p* < 0.05, Tukey’s test) (Figure 1 and Appendix A). To obtain adequate sprout suppression over long periods of time using EOs, repeated applications are often necessary on a weekly or monthly basis [21,27,28]. It is possible that longer-term and more effective sprout suppression could be achieved with *O. mixta*, *C. zeylanicum* (bark), *C. ladanifer*, *S. sclarea*, *L. nobilis*, and *A. herba-alba* EOs if repeated applications were used.

Gas chromatography (GC)—mass spectroscopy (MS)—flame ionization detection (FID) analysis of *A. herba-alba* EO revealed α-thujone, hexadecenoic acid, β-thujone, camphor, sabinene, and camphene as constituents present at amounts >1% (Table 3). Of these compounds, α-thujone made up 63.606% of the EO composition. Interestingly, thujone has previously been reported as an ineffective sprout suppressant when used alone [1]. Although camphor has been reported as a somewhat effective sprout suppressant, its relatively low concentration compared to thujone makes it unlikely that this compound alone is responsible for the sprout-suppressant properties of *A. herba-alba* EO [1]. It is possible that synergistic interactions between the various constituents within *A. herba-alba* EO may lay behind its sprout-suppressive capabilities. Further study with pure compounds and EO blends experiments could be used to test this hypothesis.

Existing chemical analyses of *O. mixta*, *C. zeylanicum* (bark), *C. ladanifer*, *S. sclarea*, *O. basilicum*, and *L. nobilis* EOs report the presence of compounds that have been previously associated with sprout suppression (Table 4). For example, 1,8-cineole, an effective sprout suppressant when used alone, can be found in the EOs of *O. mixta*, *O. basilicum* and *L. nobilis* [1,32,33,34,35]. Other effective compounds such as terpinene-4-ol and geranyl acetate can be found in *C. ladanifer* and *S. sclarea* EOs, respectively [36,37]. Furthermore, several compounds that are somewhat effective sprout suppressants including camphor and α-pinene can be found in several of the effective EOs in this study (Table 4). *Laurus nobilis* is also reported to contain α-terpineol and caryophyllene oxide [32], while *S. sclarea* contains these and citral [36]; all these compounds were reported to possess sprout-suppressive properties in the recent literature [38]. It is possible that the efficacy of these EOs as sprout suppressants can be traced to the presence of these specific compounds in their chemical profiles. GC-MS analysis of the specific EOs used in this study could support the presence of such molecules and determine their overall concentrations, although the observable effects could be significantly influenced by the relative amounts of major compounds due to various synergistic and antagonistic interactions.

Interestingly, many of the aforementioned effective compounds can be found in several of the EOs that demonstrated no significant effect on sprout length in the present study (Table 4). For example, nonanol and methyl benzoate may be present in *C. canephora* and *S. tonkinenesis* EOs, respectively, and α-pinene can be found in *C. zeylanicum* (leaf and bark), *C. atlantica*, and *A. archangelica* EOs [39,40,41,42,43]. It is possible that the concentrations of these compounds in these EOs were not high enough to achieve any noticeable effects on tuber sprouting in the present study.

### 2.2. Effects of Essential Oils on Number of Germinated Eyes

After 90 days of storage, a statistically significant two-way interaction between treatment and time was observed on sprout number (*p* < 0.001). This suggests that the impact of treatment on sprout number depends on the amount of time that has passed. Furthermore, the main effects of both treatment and time were significant (*p* < 0.001).

Treatment with *C. zeylanicum* (bark), *A. herba-alba*, and *L. nobilis* EOs resulted in a significant difference in sprout number relative to the control (Table 5). *A. herba-alba* EO treatment resulted in the lowest average sprout numbers throughout the 90-day storage period (Figure 2) and was the only treatment to cause significant differences in sprout number compared to the control throughout the entire length of the study (Table 5). Furthermore, *A. herba-alba* EO treatment limited sprout number to an average of less than 1 sprout per tuber for up to 60 days (Table 6). It is possible that repeated treatments of *A. herba-alba* EO could result in even longer suppression.

Interestingly, the EOs of *C. ladanifer*, *O. basilicum*, *O. mixta*, and *S. sclarea*, which were effective in suppressing sprout length were not effective at suppressing sprout number. It is possible that these two responses are controlled by different regulatory pathways within the tubers that are differentially affected by EO treatment. Furthermore, the presence or absence of certain constituents in these EOs, their relative amounts, or synergistic interactions between constituents may affect sprout initiation and elongation in different ways (Table 4). Further studies into the mode of action of EOs on sprout development could elucidate the hormonal pathways that are affected by EO treatment, the active ingredients in each effective EO, and whether there are synergistic interactions between compounds.

*Syzygium aromaticum* EO is the primary ingredient in the commercial sprout suppressant Biox-C. However, the present study did not find a significant effect of *S. aromaticum* EO treatment on either sprout length or sprout number (Figure 1 and Figure 2) when applied using our fumigation method. Previous studies on the active ingredient of *S. aromaticum* EO—eugenol—found the compound to be an ineffective sprout suppressant [1]. However, the continued use of Biox-C as a sprout suppressant suggests that there may be situations in which *S. aromaticum* EO can adequately suppress sprouting, perhaps depending on application scheme and cultivars used. Indeed, repeated applications of *S. aromaticum* EO is often required on a weekly to monthly basis [48]. It is possible that repeated applications of this EO could result in better sprout control. However, the present study does not demonstrate a single application of *S. aromaticum* EO as an effective sprout suppressant for cv. Ranger Russet potatoes over a 90-day storage period.

Certain limitations of the present study necessitate additional studies of *A. herba-alba* and other effective EOs as sprout suppressants prior to commercialization research. For example, the present study did not control for tuber weight. Tuber weight is an important metric as it is related to the surface area of the tubers and thereby sprouting density. Because tuber weight was not controlled for across treatment groups, it is possible that the observed effects on sprouting may be exaggerated if significantly smaller tubers on average were treated with effective EOs than those included in the control group. Nevertheless, the highly significant effect of *A. herba-alba* EO in particular warrants further investigation of this EO as a sprout suppressant. The present study also did not evaluate EO effects on tuber quality. EO exposure may alter tuber texture, reducing sugar content and flavor, or may mask internal rot of the tuber flesh. Subsequent studies are therefore needed to evaluate the impact of the effective EOs on these tuber quality measures.

Moreover, though several EOs were identified as effective sprout suppressants, additional studies at commercial or semi-commercial scales are necessary to validate these findings in an industrial setting using current application technologies. Furthermore, studies comparing these EO treatments to other commercial sprout suppressants, such as CIPC or Biox-M, in various potato cultivars could give producers and processors greater power when choosing between numerous treatments. Finally, as repeated applications of EO treatments are often required to maintain sprouting over longer periods, fine-tuning the application schedule for various cultivars will be necessary to minimize costs while maximizing control.

## 3. Materials and Methods

### 3.1. Plant Material

Potato tubers of cv. Ranger Russet were obtained from the Oregon State University Hermiston Agricultural Research and Extension Center in Hermiston, OR, USA. Tubers were harvested in September 2021 and stored in 22.5 kg mesh bags in a cooler set to 4 ºC prior to use in December 2021. Tubers were not treated with any chemicals prior to the start of the experiment.

### 3.2. Experimental Materials

A total of 21 essential oils including atlas cedarwood (*Cedrus atlantica* (Endl.) Manetti ex Carrière), celery seed (*Apium graveolens* L.), blue chamomile (*Matricaria recutita* L.), Moroccan chamomile (*Ormenis mixta* (L.) Dumort.), Roman chamomile (*Anthemis nobilis* L.), cinnamon bark (*Cinnamomum zeylanicum* Blume), cinnamon leaf (*Cinnamomum zeylanicum* Blume), cistus (*Cistus ladanifer* L.), clary sage (*Salvia sclarea* L.), clove bud (*Syzygium aromaticum* (L.) Merr. & L.M. Perry), cocoa (*Theobroma cacao* L.), coffee (*Coffea canephora* Pierre ex A.Froehner), copaiba balsam (*Copaifera langsdorfi* Desf.), allspice (*Pimenta officinalis* Lindl.), amyris (*Amyris balsamifera* L.), angelic root (*Angelica archangelica* L.), anise seed (*Pimpinella anisum* L.), armoise (*Artemisia herba-alba* Asso), basil (*Ocimum basilicum* L.), bay laurel (*Laurus nobilis* L.), and benzoin resin oil (*Styrax tonkinensis* (Pierre) Craib ex Hartwich) were used. *L. nobilis* and *C. ladanifer* EOs were purchased from Eden Botanicals. *S. aromaticum* EO was purchased from The Essential Oil Company. All other EOs were purchased from Mountain Rose Herbs.

### 3.3. Gas Chromatography Mass Spectrometry Flame Ionization Detection (GC–MS–FID) Essential Oil Analysis

Gas chromatography (GC)—mass spectroscopy (MS)—flame ionization detection (FID) analysis of *A. herba-alba* EO was performed at the Natural Products Center of the USDA-ARS, Natural Products Utilization Research Unit in University, MS, USA. Fifty μL of oil (weight also recorded) from each sample was transferred into a 10 mL volumetric flask. Samples were brought to volume with chloroform.

Oil samples were analyzed by GC–MS–FID on an Agilent (Santa Clara, CA, USA) 7890A GC system coupled to an Agilent 5975C inert XL MSD. Chemical standards and oils were analyzed using a DB-5 column (30 m × 0.25 mm fused silica capillary column, film thickness of 0.25 µm) operated using an injector temp of 240 °C, column temperature of 60 to 240 °C at 3 °C/min, and held at 240 °C for 5 min, with helium as the carrier gas, an injection volume of 1 µL (split ratio 25:1), and an MS mass range from 50 to 550. FID temperature was 300 °C. Post-column splitting was performed so that 50% of outlet flow proceeded to FID and 50% to mass spectrometry (MS) detection.

Compounds were identified by Kovats Index analyses, direct comparison of MS and retention time to authentic standards, and comparison of mass spectra with those report-ed in the Adams and NIST mass spectra databases, unless otherwise noted. Commercial standards of sabinene, α-terpinene, *p*-cymene, eucalyptol, γ-terpinene, β-thujone, (R)-(+)-camphor, terpinen-4-ol, myrtenal, cuminaldehyde, and piperitone were obtained from Sigma-Aldrich (St. Louis, MO, USA). Tricyclene was obtained from Caymen Chemicals (Ann Arbor, MI, USA). Camphene and endo-borneol were obtained from Fluka (via Sigma-Aldrich, St. Louis, MO, USA). Germacrene D, (-)-α-thujone, and an α-thujone/β-thujone mixture were obtained from Supelco (via Sigma-Aldrich, St. Louis, MO, USA). Standards were used for direct comparison with retention time and MS data providing unequivocal identification.

Compounds were quantified by performing area percentage calculations based on the total combined FID area. For example, the area for each reported peak was divided by total integrated area from the FID chromatogram from all reported peaks and multiplied by 100 to arrive at a percentage. The percentage of a peak is a percentage relative to all other constituents integrated in the FID chromatogram.

### 3.4. Experimental Design

A cotton ball was placed in the middle of a filter-paper-lined Petri dish in the center of a new black 20 L container. An amount of 1 mL of EO was pipetted on to the cotton ball. 5 randomly selected tubers were placed in the container which was then sealed with aluminum foil for fumigation with the EO vapor (Figure 3). The tubers had no direct contact with the Petri dish with EO. A lid was loosely placed on the containers which were then stacked and left undisturbed aside from scheduled intervals for data collection. There were 3 replications per EO treatment and the control with 1 mL distilled water, for a total of 3 mL of each EO and distilled water used in the experiment. The experiment was conducted at room temperature and lasted 90 days.

### 3.5. Observations

Data on sprout length and number of sprouts were collected starting at 30 days and continuing every 15 days thereafter until a 90-day storage period was reached. The longest sprout on each tuber in each replication was recorded in millimeters and reported as sprout length. The total number of germinated (≥1 mm) eyes was recorded for all tubers in each replication and reported as sprout number. Averages of observations for each replication were calculated for later analysis.

### 3.6. Statistical Analysis

R software, Version 3.6.3, was used for the statistical analysis [49]. A linear mixed model was used to analyze both sprout length and sprout number. Due to the wide variability of the data and to fulfill the ANOVA assumptions, a (log + 1) transformation was used on the sprout length and number data to achieve homogeneity of variance and normality of residuals. Significance tests were performed using chi-square tests in the “car” package in R [50]. A post hoc Tukey’s HSD test was used as a multiple comparison test to identify differences in sprout length and number due to the different treatments across all time points. Estimated marginal means and confidence intervals were back-transformed for reporting and graphics. To perform the aforementioned analysis, data summary, and graphics, we used various R packages (“tidyverse”, “ggpubr”, “rstatix”, “nlme”, “emmeans” and “ggplot2”) [51,52,53,54,55].

## 4. Conclusions

Essential oils offer a promising alternative to commonly used potato sprout suppressants including CIPC as modern regulations redefine the industry. Though several EO sprout suppressants are already commercially available, their efficacy largely depends on potato cultivar, storage temperature, and application scheme. EOs may contain numerous secondary metabolites, many of which may be found in the EOs of several other species. Moreover, the large variability in secondary metabolites found in EOs suggests that there may be other EOs with sprout suppressant capabilities that have yet to be identified. The present study demonstrates the ability of *Artemisia herba-alba* EO to significantly suppress both sprout length and number in Ranger Russet potatoes for up to 90 days at room temperature. *Cistus ladanifer*, *Ocimum basilicum*, *Ormenis mixta*, and *Salvia sclarea* EOs significantly reduced sprout length for shorter storage periods, whereas *Cinnamomum zeylanicum* (cinnamon bark), and *Laurus nobilis* EOs also significantly reduced sprout number, though to a lesser degree than *A. herba-alba* EO. It is possible that the effects of these EOs could be enhanced if they are applied repeatedly throughout storage. These results clearly show the ability of certain EOs to control potato sprouting and justify their continued investigation on a commercial scale; this could satisfy a growing need for organic alternatives in a shifting regulatory landscape and allow potato producers greater control over sprouting in their operations.

## Figures and Tables

**Figure 1 plants-11-03109-f001:**
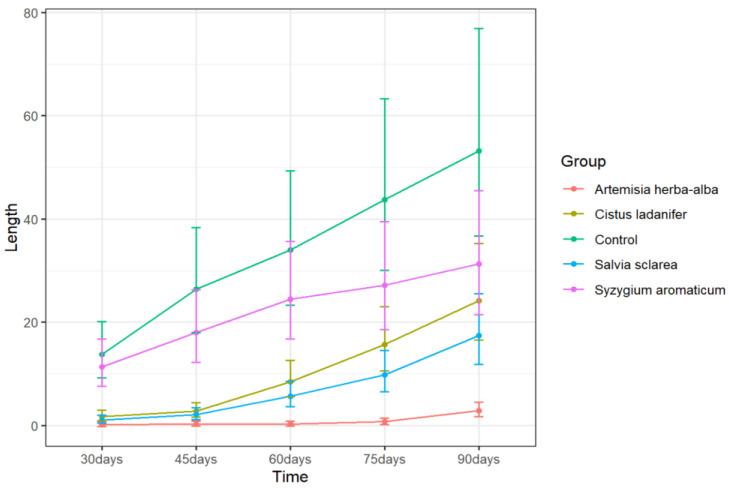
Sprout length (mm) over time of potatoes treated with distilled water (control), and the most effective EO treatments including *A. herba-alba*, *C. ladanifer*, and *S. sclarea* EOs, and *Syzygium* EO treatment. Error bars represent the 95% confidence level of the back-transformed means (*emmeans method*).

**Figure 2 plants-11-03109-f002:**
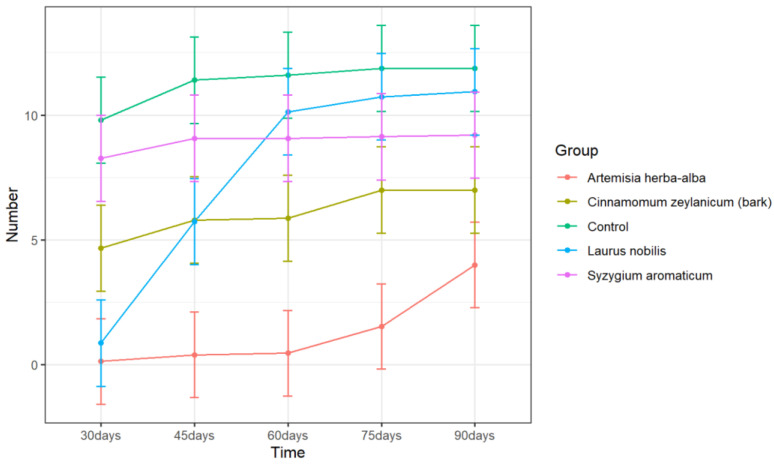
Number of sprouts per tuber over time of potatoes treated with distilled water (control), *A. herba-alba*, *C. zeylanicum* (bark), *L. nobilis*, and *S. aromaticum* EOs. Error bars represent the 95% confidence level of the means (*emmeans method*).

**Figure 3 plants-11-03109-f003:**
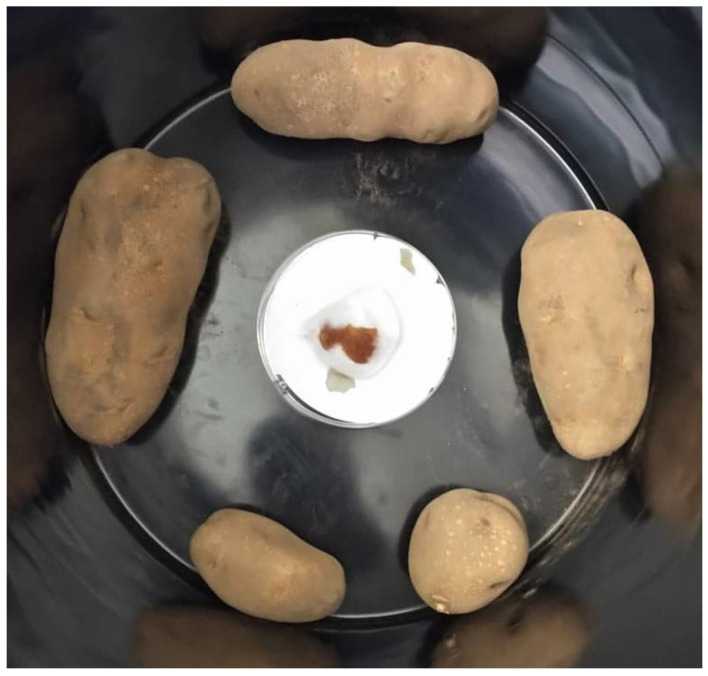
The experimental setup with tubers placed around a Petri dish lined with filter paper containing a cotton ball onto which essential oil was placed.

**Table 1 plants-11-03109-t001:** Tukey’s test *p*-values describing the Eos’ effects on sprout length relative to the control at all time points.

Treatment vs. Control	30 Days	45 Days	60 Days	75 Days	90 Days
*Cedrus atlantica*	1.000	1.000	1.000	0.999	0.999
*Apium graveolens*	1.000	1.000	1.000	1.000	1.000
*Matricaria recutita*	1.000	1.000	1.000	1.000	1.000
*Ormenis mixta*	0.002 **	0.812	0.998	0.996	0.930
*Anthemis nobilis*	0.705	0.940	1.000	1.000	0.991
*Cinnamomum zeylanicum* (bark)	<0.001 ***	0.002 **	0.445	0.472	0.284
*Cinnamomum zeylanicum* (leaf)	1.000	1.000	1.000	1.000	0.999
*Cistus ladanifer*	<0.001 ***	<0.001 ***	0.002 **	0.259	0.528
*Salvia sclarea*	<0.001 ***	<0.001 ***	<0.001 ***	0.009 **	0.066
*Syzygium aromaticum*	1.000	0.939	1.000	0.996	0.966
*Theobroma cacao*	0.936	1.000	1.000	1.000	1.000
*Coffea canephora*	1.000	1.000	1.000	1.000	1.000
*Copaifera langsdorfi*	1.000	1.000	1.000	1.000	1.000
*Pimenta officinalis*	0.995	1.000	1.000	1.000	0.993
*Amyris balsamifera*	1.000	1.000	1.000	1.000	0.997
*Angelica archangelica*	1.000	1.000	1.000	1.000	1.000
*Pimpinella anisum*	1.000	1.000	1.000	0.999	0.991
*Artemisia herba-alba*	<0.001 ***	<0.001 ***	<0.001 ***	<0.001 ***	<0.001 ***
*Ocimum basilicum*	0.002 **	0.522	0.999	0.999	0.989
*Laurus nobilis*	<0.001 ***	<0.001 ***	<0.001 ***	0.067	0.411
*Styrax tonkinensis*	0.999	1.000	1.000	1.000	1.000

**, *** = statistically significant (*p* ≤ 0.05)

**Table 2 plants-11-03109-t002:** Longest sprout length of potato tubers treated with the most effective essential oils and the control at different time points.

	30 Days	45 Days	60 Days	75 Days	90 Days
Control	13.71 ± 2.64 ^a^	26.38 ± 4.92 ^a^	33.98 ± 6.29 ^a^	43.71 ± 8.04 ^a^	53.20 ± 9.74 ^a^
*Ormenis mixta*	4.61 ± 1.00 ^b^	16.88 ± 3.21 ^ab^	23.24 ± 4.36 ^ab^	27.15 ± 5.06 ^ab^	29.99 ± 5.57 ^a^
*Cinnamomum zeylanicum* (bark)	3.99 ± 0.89 ^b^	9.46 ± 1.88 ^b^	15.69 ± 3.00 ^abc^	17.76 ± 3.37 ^ab^	21.44 ± 4.03 ^a^
*Cistus ladanifer*	1.73 ± 0.49 ^bc^	2.78 ± 0.68 ^c^	8.43 ± 1.69 ^bcde^	15.70 ± 3.00 ^ab^	24.18 ± 4.52 ^a^
*Salvia sclarea*	1.05 ± 0.36 ^cd^	2.05 ± 0.54 ^c^	5.67 ± 1.20 ^ce^	9.80 ± 1.94 ^bc^	17.40 ± 3.31 ^a^
*Artemisia herba-alba*	0.11 ± 0.20 ^de^	0.25 ± 0.22 ^d^	0.25 ± 0.22 ^f^	0.71 ± 0.30 ^d^	2.83 ± 0.69 ^b^
*Ocimum basilicum*	4.63 ± 1.01 ^b^	15.40 ± 2.94 ^ab^	23.44 ± 4.39 ^ad^	29.02 ± 5.40 ^ac^	33.17 ± 6.14 ^a^
*Laurus nobilis*	0.51 ± 0.27 ^ce^	1.13 ± 0.38 ^cd^	4.63 ± 1.01 ^e^	12.69 ± 2.46 ^ac^	22.92 ± 4.30 ^a^

Values are the back-transformed means ± SE (*emmeans method*). Different letters (a–f) within columns indicate statistically significant differences between treatments (Tukey’s test *p* < 0.05).

**Table 3 plants-11-03109-t003:** *A. herba-alba* EO constituents determined via GC—MS—FID analysis.

No.	Compound Name	Retention Time	Calculated KI	Actual KI	Identified	Area%
1	tricyclene	5.755	924	926	Kovat, NIST, Adams, Commercial Standard	0.075
2	camphene	6.465	948	954	Kovat, NIST, Adams, Commercial Standard	1.294
3	sabinene	7.152	972	975	Kovat, NIST, Adams, Commercial Standard	1.952
4	α-terpinene	8.519	1013	1017	Kovat, NIST, Adams, Commercial Standard	0.101
5	*p*-cymene	8.782	1022	1024	Kovat, NIST, Adams, Commercial Standard	0.487
6	eucalyptol	9.023	1030	1031	Kovat, NIST, Adams, Commercial Standard	0.365
7	γ-terpinene	10.013	1059	1059	Kovat, NIST, Adams, Commercial Standard	0.232
8	(-)-α-thujone	12.038	1112	1102	Kovat, NIST, Adams, Commercial Standard	63.606
9	β-thujone	12.353	1120	1114	Kovat, NIST, Adams, Commercial Standard	8.523
10	unknown	12.427	-	-	-	0.799
11	trans-pinocarveol	13.263	1142	1139	Kovat, NIST, Adams [29]	0.395
12	(R)-(+)-camphor	13.434	1146	1146	Kovat, NIST, Adams, Commercial Standard	6.87
13	sabina ketone	13.938	1158	1159	Kovat, NIST, Adams, References [29,30]	0.283
14	endo-borneol	14.378	1168	1169	Kovat, NIST, Adams, Commercial Standard	0.337
15	terpinen-4-ol	14.819	1177	1177	Kovat, NIST, Adams, Commercial Standard	0.396
16	myrtenal	15.529	1192	1195	Kovat, NIST, Adams, Commercial Standard	0.14
17	cuminaldehyde	17.405	1239	1241	Kovat, NIST, Adams, Commercial Standard	0.314
18	piperitone	18.029	1254	1252	Kovat, NIST, Adams, Commercial Standard	0.123
19	germacrene D	27.485	1481	1481	Kovat, NIST, Adams, Commercial Standard [29]	0.133
20	unknown	38.184	-	-	-	0.706
21	unknown	41.588	-	-	-	0.689
22	hexadecanoic acid	45.214	1973	1959	Kovat, NIST, Adams [31]	12.181

**Table 4 plants-11-03109-t004:** Presence of effective and somewhat effective sprout suppressant compounds as defined by Boivin et al. [1] previously reported in the EOs used in this study. EOs in bold resulted in significant differences in tuber sprouting from the control.

Essential Oil	Effective						Somewhat Effective				References
	1,8-Cineoles	Terpenin-4-ol	Geranyl Acetate	2-Phenylethanol	Nonanol	Methyl Benzoate	Camphor	α-Pinene	α-Phellandrene	Trans-Anethole	
*Cedrus atlantica*								x			[39]
** *Ormenis mixta* **	x						x				[34,44]
** *Cinnamomum zeylanicum* ** **(bark)**								x			[43]
*Cinnamomum zeylanicum* (leaf)								x			[43]
** *Cistus ladanifer* **		x						x			[37]
** *Salvia sclarea* **			x								[36]
*Theobroma cacao*				x							[45]
*Coffea canephora*					x						[40]
*Angelica archangelica*								x	x		[42]
*Pimpinella anisum*										x	[46]
** *Ocimum basilicum* **	x						x				[35]
** *Laurus nobilis* **	x	x						x			[32,47]
*Styrax tonkinensis*						x					[41]

**Table 5 plants-11-03109-t005:** Tukey’s test *p*-values describing the Eos’ effects on sprout number relative to the control at all time points.

Treatment vs. Control	30 Days	45 Days	60 Days	75 Days	90 Days
*Cedrus atlantica*	1.000	1.000	1.000	1.000	1.000
*Apium graveolens*	1.000	1.000	1.000	1.000	1.000
*Matricaria recutita*	1.000	1.000	1.000	1.000	1.000
*Ormenis mixta*	0.995	1.000	1.000	1.000	1.000
*Anthemis nobilis*	1.000	1.000	1.000	1.000	1.000
*Cinnamomum zeylanicum* (bark)	0.013 *	0.022 *	0.001 **	0.075	0.349
*Cinnamomum zeylanicum* (leaf)	1.000	1.000	1.000	1.000	1.000
*Cistus ladanifer*	1.000	1.000	1.000	1.000	1.000
*Salvia sclarea*	0.978	1.000	1.000	1.000	1.000
*Syzygium aromaticum*	1.000	0.995	0.917	0.938	0.996
*Theobroma cacao*	1.000	1.000	1.000	1.000	1.000
*Coffea canephora*	1.000	1.000	1.000	1.000	1.000
*Copaifera langsdorfi*	1.000	1.000	1.000	1.000	1.000
*Pimenta officinalis*	1.000	1.000	1.000	1.000	1.000
*Amyris balsamifera*	1.000	1.000	1.000	1.000	1.000
*Angelica archangelica*	1.000	1.000	1.000	1.000	1.000
*Pimpinella anisum*	1.000	1.000	1.000	1.000	1.000
*Artemisia herba-alba*	<0.001 ***	<0.001 ***	<0.001 ***	<0.001 ***	<0.001 ***
*Ocimum basilicum*	1.000	1.000	1.000	1.000	1.000
*Laurus nobilis*	<0.001 ***	0.007 **	1.000	1.000	1.000
*Styrax tonkinensis*	1.000	1.000	1.000	1.000	1.000

*, **, *** = statistically significant (*p* ≤ 0.05)

**Table 6 plants-11-03109-t006:** Number of germinated eyes of potato tubers treated with the most effective essential oils and the control at different time points.

	30 Days	45 Days	60 Days	75 Days	90 Days
Control	9.80 ± 0.86 ^a^	11.40 ± 0.86 ^a^	11.60 ± 0.86 ^a^	11.87 ± 0.86 ^a^	11.87 ± 0.86 ^a^
*Cinnamomum zeylanicum* (bark)	4.67 ± 0.86 ^b^	5.80 ± 0.86 ^b^	5.87 ± 0.86 ^b^	7.00 ± 0.86 ^bc^	7.00 ± 0.86 ^bc^
*Artemisia herba-alba*	0.13 ± 0.86 ^b^	0.40 ± 0.86 ^c^	0.46 ± 0.86 ^c^	1.53 ± 0.86 ^c^	4.00 ± 0.86 ^c^
*Laurus nobilis*	0.87 ± 0.86 ^b^	5.73 ± 0.86 ^b^	10.13 ± 0.86 ^ab^	10.73 ± 0.86 ^ab^	10.93 ± 0.86 ^ab^

Values are the estimated means ± SE (*emmeans method*). Different letters (a–c) indicate statistically significant differences between treatments (Tukey’s test *p* < 0.05).

## Data Availability

Not applicable.

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
