# Peer review of "Effects of Essential Oil Fumigation on Potato Sprouting at Room-Temperature Storage"

_plants, 2022, doi:10.3390/plants11223109_

Round 1

Reviewer 1 Report

This manuscript reports an experiment on “Artemisia herba-alba essential oil is a more effective potato sprout suppressant than clove and other oils at room temperature”. This is an interesting area of work. Overall, this is a  good and high-quality manuscript. The novelty and aim of the study are clear, the English style is good, the methods used are sufficient and the conclusions are supported by the results. I have minor suggestions as follows ;

·        Did the authors take pictures while conducting the experiments? Pictures could provide further evidence to support the collected data.

·        In materials and methods, it would be better to introduce GCMS analysis first (after 3.2 Experimental materials), instead of after 3. 5 Statistical Analysis.

Author Response

This manuscript reports an experiment on “Artemisia herba-alba essential oil is a more effective potato sprout suppressant than clove and other oils at room temperature”. This is an interesting area of work. Overall, this is a  good and high-quality manuscript. The novelty and aim of the study are clear, the English style is good, the methods used are sufficient and the conclusions are supported by the results. I have minor suggestions as follows ;

  • Did the authors take pictures while conducting the experiments? Pictures could provide further evidence to support the collected data.

Response: We only have a picture comparing one rep of the A. herba-alba treated potatoes to one rep of the control tubers at 60 days. These pictures have been included in the supplementary materials.

  • In materials and methods, it would be better to introduce GCMS analysis first (after 3.2 Experimental materials), instead of after 3. 5 Statistical Analysis.

Response: The section on GCMS analysis has been moved accordingly.

Reviewer 2 Report

The manuscript is interesting for the journal, however there are important aspects that must be clarified to be published.

General comments

-       The aim of the present study was to evaluate the sprout-suppressing qualities of previously untested essential oils in a single potato variety stored at room temperature, with the aim of identifying promising new sprout suppressants. So, why is the title only talking about Artemisia herba-alba essential oil and only this EO is characterized?

-       The title must be changed: (i) for a more attractive one since the proposed one does not seem like a title but a phrase in which a comparison is made, (ii) adapt the title according to the previous comment.

-       Although quite recent references are generally used, in the discussion of the results most of the citations used are prior to 2015. This is not admissible in the case of a subject that is current and on which there are many studies that have been published recently.

Specific references

LINE 84. Rename the section to “Effects of essential oils (EOs) on sprout length”

LINES 85-86. Table 1 would not be necessary if the results obtained are specified in the text.

LINE 153. Use the format for references with Boivin et al. (2020).

LINE 156. A table similar to the one made in the previous section (Table 3) is missing to see in more detail the results obtained from the number of germinated eyes.

LINE 227. It would be interesting to include a photo or diagram of the system used to treat potatoes with EOs. This would help to better understand the experimental design.

Author Response

The manuscript is interesting for the journal, however there are important aspects that must be clarified to be published.

General comments

-       The aim of the present study was to evaluate the sprout-suppressing qualities of previously untested essential oils in a single potato variety stored at room temperature, with the aim of identifying promising new sprout suppressants. So, why is the title only talking about Artemisia herba-alba essential oil and only this EO is characterized?

Response: A. herba-alba EO was by far the most effective sprout suppressant of all the EOs tested. For this reason, and for the most efficient use of time and funding, only A. herba-alba EO was characterized. To address the lack of characterization of the other EOs, the manuscript discusses existing literature describing their respective compositions. As the purpose of the study was to evaluate the effect of EOs on potato sprouting and not to elucidate the mode of action of effective EOs, we believe that this is sufficient. Future studies may clarify the specific active ingredient(s) in A. herba-alba EO or characterize the composition of the other EOs used. The title focused on A. herba-alba EO in particular as this was the most significant finding in the study. However, the title has been modified to more generally describe the study that was conducted.

-       The title must be changed: (i) for a more attractive one since the proposed one does not seem like a title but a phrase in which a comparison is made, (ii) adapt the title according to the previous comment.

Response: The title has been changed to “The effects of essential oil fumigation on potato sprouting in a single potato cultivar at room temperature storage”. This more general title captures the true purpose of the study.

-       Although quite recent references are generally used, in the discussion of the results most of the citations used are prior to 2015. This is not admissible in the case of a subject that is current and on which there are many studies that have been published recently.

Response: Many of the references [29 - 48] in the discussion section are referring to studies performing compositional analysis of the studied EOs. These specific studies were chosen as they were performed on EOs extracted from plant parts obtained from the same or nearby regions from which the EOs used in this study were reported to have originated. Because EO compositions can differ based on where the crop is grown, we feel that this justifies the use of these studies over more current ones.

Specific references

LINE 84. Rename the section to “Effects of essential oils (EOs) on sprout length”

Response: The section has been renamed accordingly.

LINES 85-86. Table 1 would not be necessary if the results obtained are specified in the text.

Response: Table 1 has been removed and its contents reported in the text (Lines 88, 90, 179, and 181). Table numbers for all subsequent tables have been adjusted accordingly.

LINE 153. Use the format for references with Boivin et al. (2020).

Response: This format for this reference has been corrected.

LINE 156. A table similar to the one made in the previous section (Table 3) is missing to see in more detail the results obtained from the number of germinated eyes.

Response: A table was created displaying detailed results pertaining to number of germinated eyes in the most effective treatments and the control (Table 6). A reference in the text to this table was inserted (Line 188).

LINE 227. It would be interesting to include a photo or diagram of the system used to treat potatoes with EOs. This would help to better understand the experimental design.

Response: Response: A pictograph of the experimental setup has been placed in section 3.4 Experimental Design in the Materials and Methods section (Figure 3, Lines 301-302).

Reviewer 3 Report

In the study, the authors report that at room temperature, the essential oil of Artemisia herba-alba is more efficient as a sprout suppressor in potatoes than clove and other oils. It is important to note that potatoes, a global staple, help battle famine. Potato sprouting causes food waste and threatens food security. Stricter laws on CIPC, the dominant chemical sprout suppressor in the last half-century, have prompted interest in biopesticides such essential oils (EOs). This study evaluated twenty-one EOs to inhibit sprouting at room temperature-stored Ranger Russet potatoes. Artemisia herba-alba EO decreased sprout length and number during 90-day storage. A. herba-alba EO contained >1% -thujone, hexadecenoic acid, -thujone, camphor, sabinene, and camphene. Cistus ladanifer, Ocimum basilicum, Ormenis mixta, and Salvia sclarea EOs shortened sprout storage durations. EOs from Cinnamomum zeylanicum (bark) and Laurus nobilis reduced sprout number. Clove oil didn't impede growth at room temperature. These results suggest adopting EOs as sprout suppressants for room-temperature potato storage, offering organic and conventional potato sectors a much-needed choice.

The study is quite interesting. However, I would like to see the following changes in the manuscript before it can be accepted.

1.       A pictograph of the experimental setup should be provided.

2.       What was the weight of the potato used?

3.       What was the age of the potato post-harvesting? Under what conditions were the potatoes stored before use?

4.       The pictures of the potatoes before and after the experiment should be provided as supplementary information.

5.       The differences in color variation and texture (puncture strength) before and after the experiment should be provided.

Author Response

In the study, the authors report that at room temperature, the essential oil of Artemisia herba-alba is more efficient as a sprout suppressor in potatoes than clove and other oils. It is important to note that potatoes, a global staple, help battle famine. Potato sprouting causes food waste and threatens food security. Stricter laws on CIPC, the dominant chemical sprout suppressor in the last half-century, have prompted interest in biopesticides such essential oils (EOs). This study evaluated twenty-one EOs to inhibit sprouting at room temperature-stored Ranger Russet potatoes. Artemisia herba-alba EO decreased sprout length and number during 90-day storage. A. herba-alba EO contained >1% -thujone, hexadecenoic acid, -thujone, camphor, sabinene, and camphene. Cistus ladanifer, Ocimum basilicum, Ormenis mixta, and Salvia sclarea EOs shortened sprout storage durations. EOs from Cinnamomum zeylanicum (bark) and Laurus nobilis reduced sprout number. Clove oil didn't impede growth at room temperature. These results suggest adopting EOs as sprout suppressants for room-temperature potato storage, offering organic and conventional potato sectors a much-needed choice.

The study is quite interesting. However, I would like to see the following changes in the manuscript before it can be accepted.

  1. A pictograph of the experimental setup should be provided.

Response: A pictograph of the experimental setup has been placed in section 3.4 Experimental Design in the Materials and Methods section (Figure 3, Lines 301-302).

  1. What was the weight of the potato used?

Response: Potato weight loss during storage is a valuable indicator of potato quality. However, potato weight was not recorded as the study focused on the effect of EOs solely on potato sprouting.

  1. What was the age of the potato post-harvesting? Under what conditions were the potatoes stored before use?

Response: Section 3.1 Plant Material of the Materials and Methods has been edited to include when the study was initiated and other storage conditions (Lines 241-242).

  1. The pictures of the potatoes before and after the experiment should be provided as supplementary information.

Response: We only have pictures comparing one rep of the A. herba-alba treated potatoes and to one rep of the control at 60 days. These pictures have been included in the supplementary materials.

  1. The differences in color variation and texture (puncture strength) before and after the experiment should be provided.

Response: Color variation and puncture strength are other good indicators of potato quality. Unfortunately, this data was not collected in the present study as it focused solely on sprout suppression due to EO fumigation. Furthermore, we are not aware of studies associating EO treatment with differences in tuber color or puncture strength.

Round 2

Reviewer 2 Report

The paper has been improved following the comments of the reviewer. However, there are some important suggestions that still need improvement to be accepted:

-       The title has been improved, however there are still aspects included in it that are not necessary. One possibility could be: “Effects of essential oil fumigation on potato sprouting stored at room temperature”

-       I understand the explanation of the authors regarding the references used to discuss the results. However, some of them are quite old (2002-2004). I suggest that at least some of them be replaced by more recent ones.

-       LINE 156. Use the format for references with Boivin et al. (2020). Thus, “as defined by [1]” should be replaced by “as defined by Boivin et al. [1]”

Author Response

The paper has been improved following the comments of the reviewer. However, there are some important suggestions that still need improvement to be accepted:

-       The title has been improved, however there are still aspects included in it that are not necessary. One possibility could be: “Effects of essential oil fumigation on potato sprouting stored at room temperature”

Response: The title has been changed to “Effects of essential oil fumigation on potato sprouting at room temperature storage”.

-       I understand the explanation of the authors regarding the references used to discuss the results. However, some of them are quite old (2002-2004). I suggest that at least some of them be replaced by more recent ones.

Response: Some of the oldest references used in the discussion have been replaced by more recent ones:

[28] American Journal of Potato Research, Kleinkopf et al. 2003 Sprout inhibition in storage… replaced with Recherche Agronomique Suisse ,Visse-Mansiaux et al. 2021 Storage of processing potato…

[36] Journal of Agricultural and Food Chemistry, Pitarokili et al. 2002 Composition and Antifungal Activity… replaced with Journal of Agronomy, Technology and Engineering Management, Aćimović et al. 2018 Salvia sclarea: chemical composition…

[43] Journal of National Science Foundation Sri Lanka, Paranagama et al. 2001 A Comparison of Essential Oil Constituents… replaced with Tropical Agricultural Research, Liyanage et al. 2017 Comparative study on major…

[48] University of Idaho Extension, Frazier et al. 2004 Organic and alternative methods… replaced with Olsen et al. 2022 Potato Sprout Suppression…

-       LINE 156. Use the format for references with Boivin et al. (2020). Thus, “as defined by [1]” should be replaced by “as defined by Boivin et al. [1]”

Response: The format for this reference has been corrected as suggested.

Reviewer 3 Report

After going through the responses of the authors, I feel there is serious flaw in their study. The study did not measure the metrics of the potatoes used. This is of utmost importance as the weight of potatoes will definitely affect the surface area of the potatoes (considering that the density of the potatoes remains within a certain limit). Hence, a change in the surface area will also affect the sprouting density.

Also, the experimental protocol reports a longer incubation period to analyze sprouting. However, the lack of texture data is surprising. The texture data can provide information on the rotting of the samples from within. On the surface, due to the activity of EO the rotting may not happen. But the interior, where the availability of EO is scant, there may be rotting of the tuber. Authors have overseen this information.

Author Response

After going through the responses of the authors, I feel there is serious flaw in their study. The study did not measure the metrics of the potatoes used. This is of utmost importance as the weight of potatoes will definitely affect the surface area of the potatoes (considering that the density of the potatoes remains within a certain limit). Hence, a change in the surface area will also affect the sprouting density.

Response: The authors acknowledge the importance of collecting tuber weight data going forward. However, we do not believe that the lack of this metric invalidates the most important result of the present study. That is, the highly significant result of A. herba-alba EO as a sprout suppressant. Sprout length was limited to 1 mm for up to 75 days, and 3 mm for up to 90 days with A. herba-alba EO compared to the control which measured 43 mm and 53 mm at 75 and 90 days, respectively. Similarly, sprout number was limited to 1 sprout per tuber for up to 60 days with A. herba-alba EO treatment compared to the control which exhibited >10 sprouts per tuber. While it is possible that the observed effect of A. herba-alba may not have been as strong if potato weight had been controlled for, the authors believe that the observed effects are significant enough to merit publication and further investigation. The limitation of not controlling for tuber weight has been incorporated into the discussion (Lines 235 - 244).

Also, the experimental protocol reports a longer incubation period to analyze sprouting. However, the lack of texture data is surprising. The texture data can provide information on the rotting of the samples from within. On the surface, due to the activity of EO the rotting may not happen. But the interior, where the availability of EO is scant, there may be rotting of the tuber. Authors have overseen this information.

Response: The authors understand the importance of tuber quality studies. However, the focus of the present study was to evaluate the effect of EOs on tuber sprouting, with the goal of identifying the most promising EO sprout suppressants. It is common knowledge that when potato tubers begin sprouting, tuber weight decreases and their marketability is significantly reduced. We are excited to report that the A. herba-alba essential oil suppresses sprouting at room temperature; this will be the first report on that essential oil. Subsequent studies of the identified EOs with sprout suppression activity may focus on the impact that these effective EOs have on tuber quality measures including tuber texture, reducing sugar content, and flavor. Also, we will be conducting further study with A. herba-alba and other EOs we identified to reveal their mode of action. Nevertheless, the reviewer’s points have been incorporated into the discussion as potential limitations of the study (Lines 244 - 247).

Round 3

Reviewer 3 Report

May be published